# Component Combination Test to Investigate Improvement of the IHACRES and GR4J Rainfall–Runoff Models

**Mun-Ju Shin** [1] **and Chung-Soo Kim** [2,*]

1    Water Resources Research Team, Jeju Province Development Corporation, 1717-35, Namjo-ro, Jocheon-eup, Jeju-si 63345, Jeju-do, Korea; mj.shin@hotmail.com
2    Department of Land, Water and Environment Research, Korea Institute of Civil Engineering and Building Technology, 283, Goyangdae-ro, Ilsanseo-gu, Goyang-si 10223, Gyeonggi-do, Korea
*    Correspondence: alska710@kict.re.kr; Tel.: +82-(0)31-9100-274

**Abstract:** Rainfall–runoff models are not perfect, and the suitability of a model structure depends on catchment characteristics and data. It is important to investigate the pros and cons of a rainfall–runoff model to improve both its high- and low-flow simulation. The production and routing components of the GR4J and IHACRES models were combined to create two new models. Specifically, the GR_IH model is the combination of the production store of the GR4J model and the routing store of the IHACRES model (vice versa in the IH_GR model). The performances of the new models were compared to those of the GR4J and IHACRES models to determine components improving the performance of the two original models. The suitability of the parameters was investigated with sensitivity analysis using 40 years' worth of spatiotemporally different data for five catchments in Australia. These five catchments consist of two wet catchments, one intermediate catchment, and two dry catchments. As a result, the effective rainfall production and routing components of the IHACRES model were most suitable for high-flow simulation of wet catchments, and the routing component improved the low-flow simulation of intermediate and one dry catchments. Both effective rainfall production and routing components of the GR4J model were suitable for low-flow simulation of one dry catchment. The routing component of the GR4J model improved the low- and high-flow simulation of wet and dry catchments, respectively, and the effective rainfall production component improved both the high- and low-flow simulations of the intermediate catchment relative to the IHACRES model. This study provides useful information for the improvement of the two models.

**Keywords:** combination of model components; rainfall–runoff model; sensitivity analysis; model improvement

## 1. Introduction

Rainfall–runoff models are used in various studies to understand and predict hydrological phenomena. The GR4J [1] and IHACRES [2] conceptual rainfall–runoff models are well known and practically used for various studies due to resulting good model performances with relatively fewer parametric and structural uncertainties [3]. Rainfall–runoff models do not perfectly simulate the observed flow [4], and the appropriate rainfall–runoff model structure depends on the catchment characteristics and simulation periods such as flood and drought periods. For example, Shin et al. [5] analyzed the adequacy of the simulation capability of four rainfall–runoff models with different complexities (4–13 parameters) for five catchments in Australia using global sensitivity analysis methods. As a result, the IHACRES model is strong for high-flow simulation, whereas the GR4J model is strong for low-flow simulation [5]. These findings were also confirmed in the analysis of the effect of time-step data on the simulation results [6] or the hydrological modeling of catchments with various geological characteristics [7]. Therefore, improving their hydrological model structures to be able to capture both high- and low-flow simulations is important.

Modelers work toward reasonable simulations of both high and low flows by improving the model structures, parameterization and/or calibration approaches [4]. Therefore, many studies investigating the parameters and model structures of rainfall–runoff models have been carried out to improve the models: uncertainty analysis of the parameters and simulation results [8]; assessment of the strengths and weaknesses of the model structure and parameters [9]; application of various uncertainty-screening methods including global sensitivity methods to examine the parametric and structural uncertainty [3,5]. However, these studies focused on the investigation of the uncertainty in the parameters and model structures. In addition, even if the hybrid objective function using the average of the high- and low-flow objective functions was used, the high- and low-flow simulations could not be improved at the same time [5]. Therefore, the methods are not sufficient for the improvement of the rainfall–runoff model structure, as they may in fact improve the results of the rainfall–runoff modelling process.

Hence, studies on the combination of components of different model structures were performed to determine pragmatic measures for the improvement of the rainfall–runoff model. For example, Clark et al. [10] developed the Framework for Understanding Structural Errors (FUSE) method, which can be used to choose suitable model structures by component combination (e.g., evaporation, percolation, and base flow) and to quantify the uncertainty of the model structures. Bai et al. [11] used the fuzzy model selection framework for the automatic generation of suitable rainfall–runoff models for the catchment databases by combining evapotranspiration, soil moisture accounting, and routing components. Fenicia et al. [12] used the SUPERFLEX framework to generate and test new rainfall–runoff models by combining three elements of the reservoir, lag function, and junction. Van Hoey et al. [13] implemented qualitative model structure sensitivity analysis by varying one model component at a time for better model selection. However, these studies focused on the choice of the best model structure; therefore, additional research is required to improve the existing rainfall–runoff model structures.

The purpose of this study was to investigate how the components of a hydrological model improve the runoff simulation results. The novelty of this study was to evaluate the strengths and weaknesses of each component through the combination of components of the existing hydrological models. In this study, the components of two existing rainfall–runoff models, GR4J [1] and IHACRES [2], were combined to create new models and the performance of the new models was compared with that of the existing models to determine how to improve the existing models. In addition, we investigated the sensitivity of the parameters of the new models using global sensitivity analysis. We investigated if the parameters of the new models are identifiable, how a parameter affects sensitivities of other parameters, why the performance of a new model is poor for a certain case, and how the rainfall–runoff models can be improved by the sensitivity analysis. It should be noted that Shin et al. [5] analyzed the adequacy of the rainfall–runoff model using only parametric sensitivity analysis. This study differs from the Shin et al. [5] study in that it analyzes the appropriateness of the rainfall–runoff model using a combination of components of the rainfall–runoff models and a sensitivity analysis method.

## 2. Materials and Methods

### 2.1. Catchment and Data

We selected five catchments of the Australian Capital Territory (ACT) in Australia as study catchments: Goodradigbee River at Brindabella (Brindabella; 427 km$^2$), Cotter River at Gingera (Gingera; 148 km$^2$), Orroral River at Crossing (Orroral Crossing; 90 km$^2$), Queanbeyan River at Tinderry (Tinderry; 490 km$^2$), and Molonglo River at Burbong (Burbong; 505 km$^2$). The four catchments except for Burbong are mountainous catchments and these catchments were selected due to having good quality and a long period of data. These catchments have different hydroclimatological conditions from wet to dry and different ranges of elevations and catchment characteristics [5]. Their high flow from the catchment has more peaks, there is a lower amount of baseflow, the runoff

coefficients are smaller, the dry periods are longer, and the flow pattern is more variable than that of the flow of European and North American catchments [14]. The Brindabella and Gingera catchments located in the western part of the ACT are wet catchments (1127 and 985 mm/yr of mean annual rainfall, and 0.31 and 0.29 of mean annual runoff ratio, respectively), whereas the Orroral Crossing catchment located in the center of the ACT is an intermediate catchment (885 mm/yr of mean annual rainfall and 0.13 of mean annual runoff ratio) and the Tinderry and Burbong catchments located in the eastern part of the ACT are dry catchments (716 and 664 mm/yr of mean annual rainfall, and 0.18 and 0.11 of mean annual runoff ratio, respectively). The different characteristics of the catchments can be useful to investigate the pros and cons of the rainfall–runoff models because the performance of a rainfall–runoff model can vary depending on the characteristics of the catchment [15]. The five catchments are described in detail in the work of Shin et al. [5].

More than 40 years' worth of climate and flow data are available for the five catchments, which have spatiotemporally varying characteristics from wet to dry, including the millennium drought [16]. We selected the daily rainfall and potential evapotranspiration data for the period from 1970 to 2009 as the input for the rainfall–runoff models. The observed daily flow data for the 40 years were used for the calibration and validation of the parameters of the rainfall–runoff models. The 40 years' worth of data were divided into four 10-year periods for the split-sampling test [17]. The calibration of the parameters was performed for each 10-year period and the validation was performed for the three other 10-year periods of the respective 10-year calibration. The duration of the calibration period was chosen based on the studies of Yapo et al. [18], Anctil et al. [19], and Kim et al. [20]. The year preceding each decade was used as warm-up period. A summary of the average rainfall and average runoff over the four decades of the five catchments is shown in Table 1.

**Table 1.** Average rainfall and runoff over four decades of five catchments.

| Catchment | Average Rainfall (mm/day) | | | | Average Runoff (mm/day) | | | |
|---|---|---|---|---|---|---|---|---|
| | 1970s | 1980s | 1990s | 2000s | 1970s | 1980s | 1990s | 2000s |
| Brindabella | 3.22 | 3.23 | 3.27 | 2.63 | 1.15 | 1.05 | 1.02 | 0.62 |
| Gingera | 2.90 | 2.77 | 2.86 | 2.29 | 1.02 | 0.80 | 0.80 | 0.46 |
| Orroral Crossing | 2.67 | 2.55 | 2.61 | 2.05 | 0.42 | 0.37 | 0.37 | 0.14 |
| Tinderry | 2.30 | 2.08 | 1.93 | 1.57 | 0.66 | 0.38 | 0.28 | 0.06 |
| Burbong | 1.96 | 1.87 | 1.93 | 1.50 | 0.35 | 0.22 | 0.19 | 0.02 |

*2.2. Calibration Method*

For the calibration of the parameters of the rainfall–runoff models, we selected the shuffled complex evolution (SCE) global optimization method [21,22], which is one of the popular parameter optimization methods [23–26]. The SCE algorithm incorporates the strengths of the simplex procedure [27], competitive evolution [28], controlled random search [29], and the concept of complex shuffling [21]. The SCE algorithm randomly chooses the initial population from the feasible parameter space. It then divides the population into complexes, which are communities with parents for the generation of offspring. The respective complex evolves independently using a competitive complex evolution strategy. Lastly, this algorithm shuffles the evolved complexes into a single population to share the information. The aforementioned steps are iterated until the results meet the preselected convergence criteria that is either the relative tolerance of 1e-5 or maximum number of iterations of 10,000 in this study. We estimated an optimal parameter set for each calibration period and rainfall–runoff model using the SCE algorithm and two objective functions of Nash–Sutcliffe efficiency (NSE) [30] and the NSE with log-transformed data (NSElog). The two objective functions will be described in Section 2.4.

### 2.3. Sensitivity Analysis Method

We employed the Sobol' method [31] for the sensitivity analysis of the parameters, which is a variance-based quantitative global sensitivity analysis method. The usefulness of this method was demonstrated in various studies (e.g., [32–34]). This method quantifies the relative contributions of the individual parameters and parameter interactions to the model output variance by the decomposition of the output variance. The Sobol' method calculates two indices, the first-order sensitivity index (FSI) and total sensitivity index (TSI), and the sensitivity indices range from 0 to 1. The FSI calculates the sensitivity of each parameter on the output, whereas the TSI calculates the sensitivity of each parameter and its interaction with the other parameters on the output [35]. The TSI provides more reliable results in the investigation of the overall effect of each parameter on the output than the FSI [35], therefore, the TSI was employed in this study. The TSI is defined as below [36]:

$$\text{TSI} = \frac{E_{X_{\sim i}}\left(V_{X_i}(Y|X_{\sim i})\right)}{V(Y)} \tag{1}$$

where $X_i$ is the $i$th parameter and $X_{\sim i}$ represents the vector of all parameters, except for $X_i$. The inner variance in the numerator means that the variance of $Y$, which is the scalar objective function values, takes into account all possible values of $X_i$ while keeping $X_{\sim i}$ fixed. The outer expectation operator in the numerator takes into account all possible values of $X_{\sim i}$ and the variance $V(Y)$ in the denominator represents the total unconditioned variance. Therefore, the TSI can be interpreted as the expected variance that would be left if all parameters are fixed, not including $X_i$ [36].

The Saltelli's scheme [37], which calculates the TSI using the reduced number of samples from $n(2k + 2)$ to $n(k + 2)$, was employed for the sensitivity analysis, where $n$ represents the initial sample size, which is 10,000 in this study, and $k$ is the number of parameters of the rainfall–runoff models. For example, the total number of samples is 60,000 for the four-parameter rainfall–runoff models. We used this scheme in the R "sensitivity" package [38] and more details about the sampling method can be found in Shin et al. [5].

### 2.4. Objective and Target Functions

Two objective functions focusing on fitting high and low flows were used for parameter calibration. The NSE, which relatively emphasizes high-flow fitting, is as follows.

$$\text{NSE} = 1 - \frac{\sum_{i=1}^{n}(Q_{obs,i} - Q_{sim,i})^2}{\sum_{i=1}^{n}\left(Q_{obs,i} - \overline{Q_{obs}}\right)^2} \tag{2}$$

where $Q_{obs,i}$ is the observed flow at the time step $i$ (daily here), $\overline{Q_{obs}}$ is the mean of the observed flow, $Q_{sim,i}$ is the simulated flow, and $n$ is the number of time steps. The NSE has the range of $[-\infty, 1]$, where a value of 1 means that the simulated flow and the observed flow are in perfect agreement. It emphasizes fitting high flow because it squares the difference between the observed and the simulated values [4,39,40]. The NSE with log-transformed data (NSElog) was used to fit low flow. The NSElog emphasizes fitting low flow because of the log transformation of time series data [4,40].

For the sensitivity analysis, we used two target functions for high and low flow, which were taken from the work of Shin et al. [5]. The NSE* [41] target function was used for parameter sensitivity analysis for high flow.

$$\text{NSE}^* = \left\{ 1 - \frac{\sum_{i=1}^{n}(Q_{obs,i} - Q_{sim,i})^2}{\sum_{i=1}^{n}\left(Q_{obs,i} - \overline{Q_{obs}}\right)^2} \right\} \Big/ \left\{ 1 + \frac{\sum_{i=1}^{n}(Q_{obs,i} - Q_{sim,i})^2}{\sum_{i=1}^{n}\left(Q_{obs,i} - \overline{Q_{obs}}\right)^2} \right\} \tag{3}$$

The NSE* divides the NSE by the NSE with a plus sign and has a range of $[-1, 1]$ rather than $[-\infty, 1]$ from NSE. This target function has the advantage of reducing the

influence of large negative NSE values, which could bias the sensitivity results, without changing the interpretation of the NSE value. The NSElog* target function, which is the NSE* with log-transformed data, was used for parameter sensitivity analysis for low flow.

*2.5. Rainfall–Runoff Models*

We used the GR4J and IHACRES models, which are well known and widely used daily conceptual rainfall–runoff models (e.g., [42,43]) to generate new models by component combination. The GR4J model has four parameters to simulate stream flow. This model has two stores (effective rainfall production and routing stores) and two unit hydrographs. The production store calculates the effective rainfall, evapotranspiration, and percolation of surface soil using the parameter $x1$ (Table 2). The routing store divides the effective rainfall into 90% and 10% to generate the quick and slow flow using the parameters $x2$, $x3$, and $x4$ (Table 2). The 90% of the effective rainfall is converted into slow flow using one unit hydrograph and the remaining 10 % of the effective rainfall is converted into quick flow using another unit hydrograph [44]. The ranges of the GR4J parameters were taken from Shin et al. [5].

**Table 2.** Description of the parameters of four rainfall–runoff models.

| Parameter | Parameter Number | Unit | Range | Description |
|:---:|:---:|:---:|:---:|:---:|
| **GR4J** | | | | |
| $x1$ | 1 | [mm] | 50–5000 | Maximum capacity of the production store |
| $x2$ | 2 | [mm] | −15 to 4 | Groundwater exchange coefficient |
| $x3$ | 3 | [mm] | 10–1300 | One day ahead maximum capacity of the routing store |
| $x4$ | 4 | [day] | 0.5–5 | Time base of unit hydrograph $UH1$ |
| **IHACRES-CMD** | | | | |
| $f$ | 1 | [–] | 0.5–1.3 | CMD stress threshold as a proportion of $d$ |
| $e$ | - | [–] | 1 (fixed) | Temperature to potential evapotranspiration (PET) conversion factor |
| $d$ | - | [mm] | 200 (fixed) | CMD threshold for producing flow |
| $\tau_s$ (tau_s) | 2 | [day] | 10–1000 | Time constant for slow-flow store |
| $\tau_q$ (tau_q) | 3 | [day] | 0–10 | Time constant for quick-flow store |
| $v_s$ (v_s) | 4 | [–] | 0–1 | Fractional volume of slow flow |
| **GR_IH** | | | | |
| $x1$ | 1 | [mm] | 50–5000 | Maximum capacity of the production store |
| $\tau_s$ (tau_s) | 2 | [day] | 10–1000 | Time constant for slow-flow store |
| $\tau_q$ (tau_q) | 3 | [day] | 0–10 | Time constant for quick-flow store |
| $v_s$ (v_s) | 4 | [–] | 0–1 | Fractional volume of slow flow |
| **IH_GR** | | | | |
| $f$ | 1 | [–] | 0.5–1.3 | CMD stress threshold as a proportion of $d$ |
| $e$ | - | [–] | 1 (fixed) | Temperature to potential evapotranspiration (PET) conversion factor |
| $d$ | - | [mm] | 200 (fixed) | CMD threshold for producing flow |
| $x2$ | 2 | [mm] | −15 to 4 | Groundwater exchange coefficient |
| $x3$ | 3 | [mm] | 10–1300 | One day ahead maximum capacity of the routing store |
| $x4$ | 4 | [day] | 0.5–5 | Time base of unit hydrograph $UH1$ |

The IHACRES model of the catchment moisture deficit (CMD) version, which was applied in this study, accounts for the variation of the catchment moisture at each time step. This model consists of effective rainfall production and routing stores. The production store calculates the effective rainfall by combining the CMD output estimated by the accounting equation and water output, which is converted from the raw rainfall using a nonlinear function. The routing store uses the parallel storages and unit hydrographs to convert the effective rainfall into quick and slow flows. The total flow is the sum of the quick and slow flows. The parameter $e$ was fixed as unity (Table 2) because of the use of potential evapotranspiration data instead of temperature data and the parameter $d$ was fixed as

200 based on the study of Croke and Jakeman [2]. Therefore, four out of six parameters were used for the calibration: parameter $f$ is related to the production store and the parameters $\tau_s$, $\tau_q$, and $v_s$ are related to the routing store.

We created two new models by combining the effective rainfall production and routing stores of the GR4J and IHACRES models. Specifically, the GR_IH model is the combination of the production store of the GR4J model and the routing store of the IHACRES model (vice versa in the IH_GR model). The GR_IH and IH_GR models are used to investigate the improvement of the high-flow simulation of the GR4J model and the low-flow simulation of the IHACRES model, respectively. Table 2 describes the parameters of the two original rainfall–runoff models and the two new models. The two original models are included in the Hydrological Model Assessment and Development (Hydromad) package [45], which is an open source, R-based, software package (available at http://hydromad.catchment.org, accessed on 31 July 2021).

## 3. Research Procedure

Firstly, the parameters of the four rainfall–runoff models were calibrated for the four decadal datasets of the five catchments by the SCE algorithm with the NSE and NSElog objective functions. The NSE and NSElog objective function values for the calibration and validation periods of the four rainfall–runoff models were compared to determine the component(s) (i.e., the production and routing stores) that can improve the performances of the original rainfall–runoff models.

The model performance statistics such as the NSE and NSElog are statistical measures that may conceal good or bad performances for the different parts of hydrograph; therefore, the simulated flow time series should be compared to the observed flow time series to determine how each component improves the simulated flow. We extracted the hydrographs for the important periods (i.e., relatively large rainfall events) and compared the simulation results of the four rainfall–runoff models. In addition, we used flow duration curves to investigate recession- and low-flow simulations. The hydrograph and flow duration curve analyses complement the results of the statistical analysis using the model performance statistics.

Lastly, we performed the sensitivity analysis of the model parameters to establish which parameters are sensitive to the target function(s). In addition, the sensitivity analysis can be used to identify if the respective parameters of the new models generated by component combination are suitable. The sensitivity analysis complements the results of the model performance statistics.

## 4. Results and Discussion

### 4.1. Comparison of Model Performances for High-Flow Objective Function

Figure 1 shows the NSE values of the four rainfall–runoff models for the five catchments. The subtitle for each catchment (e.g., 1970s) represents the calibration period. The cross in the figure represents the calibration period and the circle represents the validation period. The simulation periods are shown in the horizontal axis and the NSE values are shown in the vertical axis. To show the results clearly, we showed the NSE values more than 0.5, which is a threshold for good model performance from the study of Moriasi et al. [46]. The NSE value ranging from 0 to 0.5 is represented as a disconnected line and the negative NSE value is not shown to distinguish the NSE value in the range between 0 and 0.5. Figure 2 shows the simulated and observed hydrographs of the maximum events in the selected decadal simulation periods for the five catchments to complement the analysis of the model performances for Figure 1. We showed the representative hydrographs in Figure 2 that can show the reason for the difference between NSE values in Figure 1. The model performances are different due to using inputs with different climate forcing and catchment characteristics.

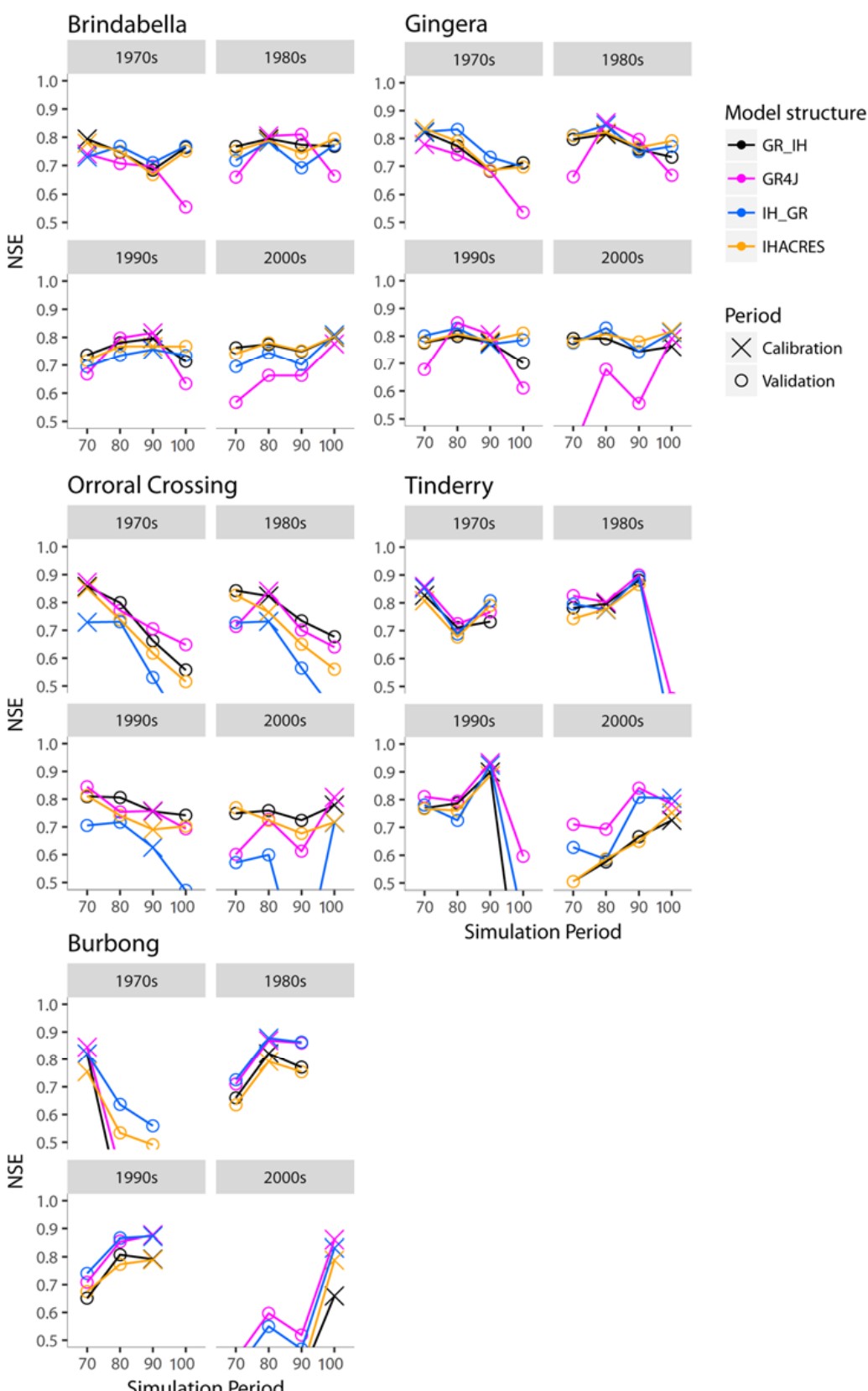

**Figure 1.** Comparison of the NSE values of the four rainfall–runoff models for the five catchments: Brindabella (wet), Gingera (wet), Orroral Crossing (intermediate), Tinderry (dry), and Burbong (dry).

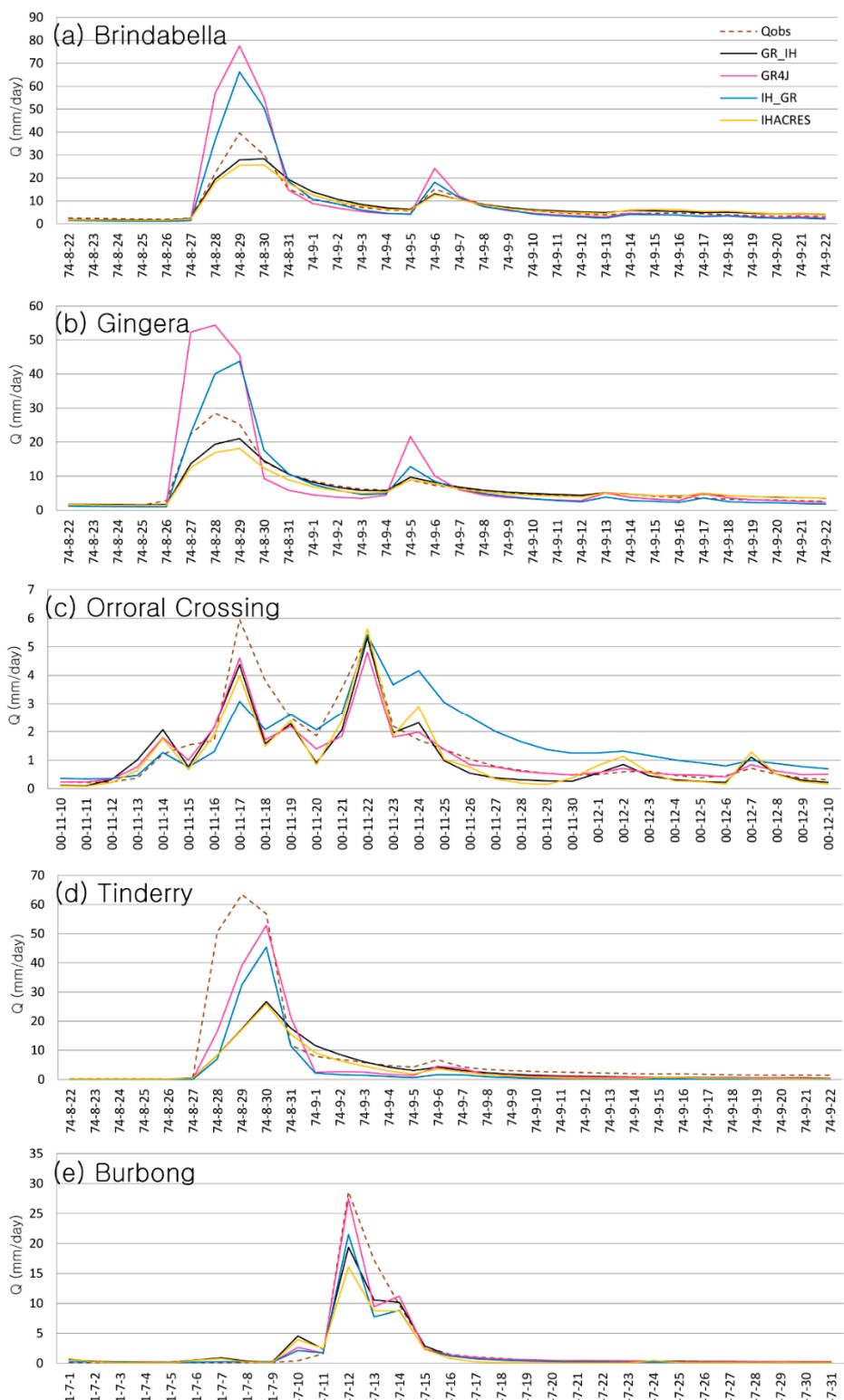

**Figure 2.** Comparison of the hydrographs of the four rainfall–runoff models for the maximum events with respect to the NSE objective function: (**a**) validation of the 1970s with the calibration of the 2000s for the Brindabella (wet) catchment, (**b**) validation of the 1970s with the calibration of the 2000s for the Gingera (wet) catchment, (**c**) validation of the 2000s with the calibration of the 1970s for the Orroral Crossing (intermediate) catchment, (**d**) validation of the 1970s with the calibration of the 2000s for the Tinderry (dry) catchment, and (**e**) the validation of the 1990s with the calibration of the 1980s for the Burbong (dry) catchment.

### 4.1.1. Wet Catchments

The GR_IH model of the Brindabella and Gingera catchments had higher NSE values overall for the validation periods compared to the GR4J model (Figure 1), especially for the validation periods with the calibration of the 2000s. The hydrographs in Figure 2a,b complement these results because the difference between the simulated and observed (Qobs) flows of the peak and recession flow was much smaller for the GR_IH model compared to the GR4J model. This confirms that the routing component of the IHACRES model predicts the high flow of the wet catchments better.

The IH_GR model had higher NSE values than the GR4J model for the validation periods with the calibration of the 2000s (Figure 1), which indicates that the production component of the IHACRES model improves the GR4J model for the high-flow simulation of the wet catchments. In addition, the GR4J model had significantly lower NSE values than the other models for the validation periods with the calibration of the dry 2000s, which indicates that the calibrated parameter values in the production store of the GR4J model for the dry period causes high-flow prediction problems for the wet periods. Therefore, an improvement of the production store parameters of the GR4J model is needed in the future. Overall, the production component of the IHACRES model improved the GR4J model with respect to the NSE values for the validation periods with the other calibration periods. However, the GR4J model had higher NSE values than the IH_GR model for the validation of the 1990s with the calibration of the 1980s, and the validation of the 1980s with the calibration of the 1990s, which indicates that the production component of the IHACRES model does not always improve the GR4J model for the high-flow simulation of the wet catchments. The hydrographs of the GR4J model in Figure 2a,b also show that the production component of the GR4J model is not suitable compared to the IH_GR model. The IHACRES model is suitable for the high-flow simulation of the wet catchments because it has the highest NSE values overall.

### 4.1.2. Intermediate Catchment

The GR_IH model of the Orroral Crossing catchment had higher NSE values for the validation periods with the calibrations of the 1980s and 2000s than the GR4J model (Figure 1), whereas it had lower NSE values for the validation of the 1990s and 2000s with the calibration of the 1970s. This means that the routing component of the IHACRES model does not always improve the performance of the GR4J model. It also implies that the data from the calibration periods such as dry and wet periods affect the improvement of the model performance in the validation period(s). The hydrograph of the GR_IH model (Figure 2c) complements this result because it shows a larger difference between simulated and observed recession flow for the GR_IH model compared to the GR4J model. However, the NSE values of the GR_IH model were higher than those of the IHACRES model; therefore, the production component of the GR4J model improves the performance of the IHACRES model.

The NSE values of the IH_GR model were lower than those of the GR4J model (Figure 1), which contrasts the results for the wet catchments. This implies that the improvement of the GR4J model performance by the production component of the IHACRES model depends on the catchment characteristics. Therefore, information of catchment characteristics such as wet and dry catchments should be considered for the improvement of the rainfall–runoff model. In addition, the hydrograph of the IH_GR model has a different shape than those of the other models (Figure 2c; i.e., smaller first peak flow and larger high flow after the second peak flow for the IH_GR model) and reveals that the combination of the production component of the IHACRES model and routing component of the GR4J model is not suitable for this catchment. The GR4J model was usually better for the high-flow simulation of the intermediate catchment than the IHACRES model with respect to the NSE values.

### 4.1.3. Dry Catchments

All rainfall–runoff models performed well for the Tinderry catchment for the simulation periods from the 1970s to 1990s except for the case of the calibration of the 2000s. The NSE values of the GR_IH model for the Tinderry and Burbong catchments were similar or lower than those of the GR4J model (Figure 1). The NSE values of the GR_IH model were always lower than those of the GR4J model with the calibration of the 2000s for the Tinderry catchment and for all cases of the Burbong catchment, which means that the routing component of the IHACRES model does not improve the high-flow simulations of the GR4J model for the dry catchments. The peak-flow simulations of the GR_IH model (Figure 2d,e) complement the results because they show a larger difference between the simulated and observed flow than the GR4J model. The NSE values and hydrographs of the GR_IH model were similar to those of the IHACRES model (Figures 1 and 2d,e), which implies that the improvement of the IHACERS model by the production component of the GR4J model is minor. Therefore, the routing component of the IHACRES model and the production component of the GR4J model are not useful for the significant improvement of high-flow simulation for the dry catchments.

The NSE values of the IH_GR model were similar or lower than those of the GR4J model (Figure 1), except for the results with the calibration of the 1970s for the Burbong catchment. Therefore, the production component of the IHACRES model did not improve the high-flow simulation of the GR4J model, and the hydrographs shown in Figure 2d,e complement this result. The IH_GR model for the Tinderry catchment had similar or higher NSE values compared to the IHACRES model and that for the Burbong catchment always had higher NSE values than the IHACRES model (Figure 1). This means that the routing component of the GR4J model improves the IHACRES model simulation for the dry catchments (Figure 2d,e); therefore, the routing component of the GR4J model is appropriate for the simulation of the dry catchments.

### 4.2. Comparison of Model Performances for Low-Flow Objective Function

Figure 3 shows the NSElog values of the four rainfall–runoff models for the five catchments. This figure also shows the performance statistics larger than 0.5 similar to Figure 1. Figure 4 shows the simulated and observed hydrographs for the maximum events in the selected decadal simulation periods of the five catchments to complement the analysis of the model performances in Figure 3. Figure 5 shows the flow duration curves for the selected decadal simulation periods in Figure 4 to investigate the recession- and low-flow simulation by the four rainfall–runoff models. Figure 5 can also complement the analysis of the model performances in Figure 3.

### 4.2.1. Wet Catchments

All rainfall–runoff models had appropriate NSElog values for the Brindabella catchment. The GR_IH model of the Brindabella and Gingera catchments had similar or lower NSElog values than the other models (Figure 3). This means that the production component of the GR4J model and routing component of the IHACRES model do not improve the performances of the IHACRES and GR4J models, respectively. Figure 4a,b also indicate the poor recession- and low-flow simulations of the GR_IH model; therefore, the model structure is not appropriate. Figure 5a shows a reasonable very low-flow simulation of the GR_IH model for the range of 50–90% flow. However, the simulation of the GR_IH model for the range of 10–40% flow, which includes the recession- and low-flow simulations, was poor and the poor simulation for this range resulted in the low NSElog value. Figure 5b shows that the simulations of the GR_IH model for low flow (20–40%) and very low flow (>50%) were poor, which complements the result in Figure 3. The reason for the poor simulation will be discussed in Section 4.3.

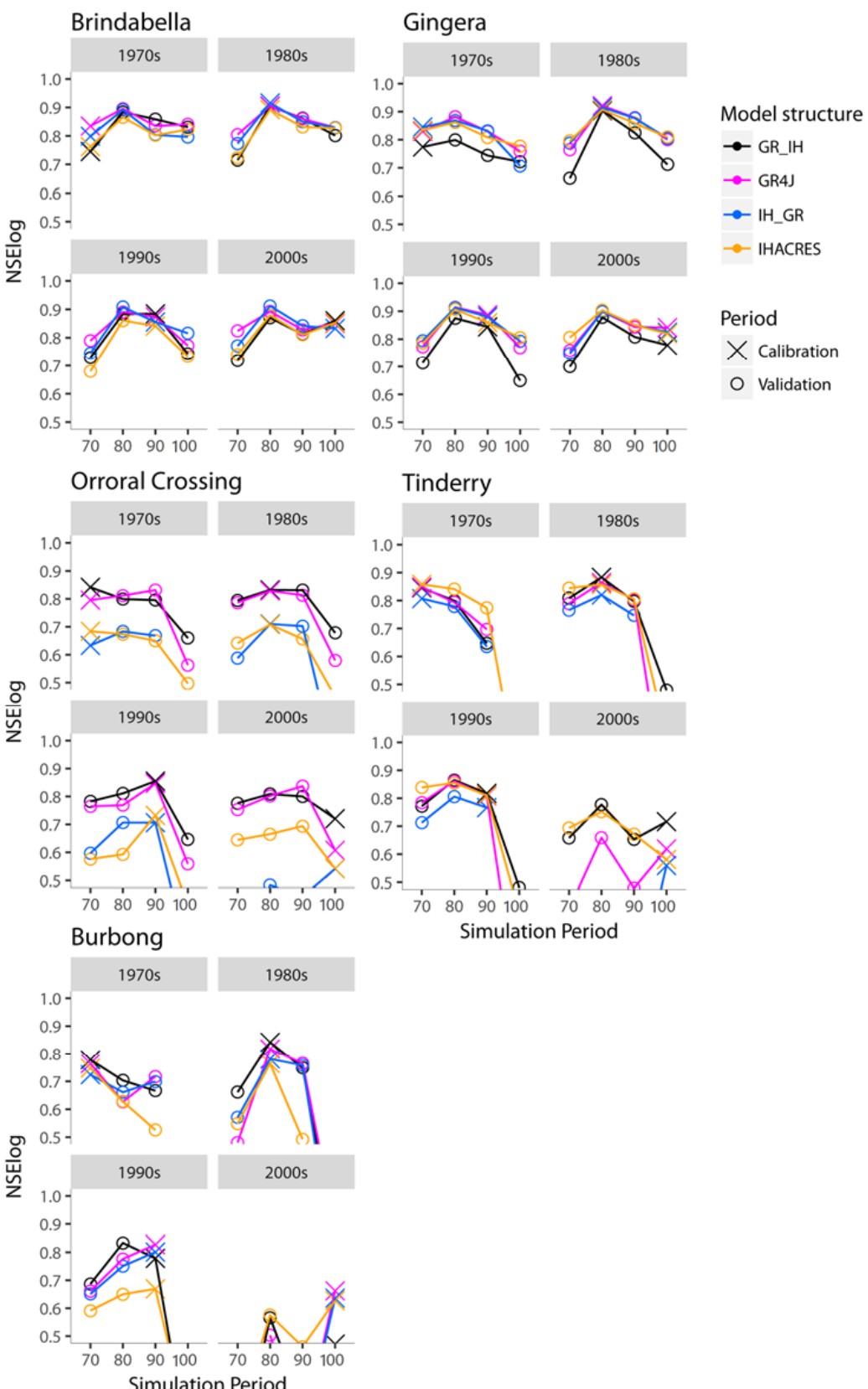

**Figure 3.** Comparison of the NSElog values of the four rainfall–runoff models for the five catchments: Brindabella (wet), Gingera (wet), Orroral Crossing (intermediate), Tinderry (dry), and Burbong (dry).

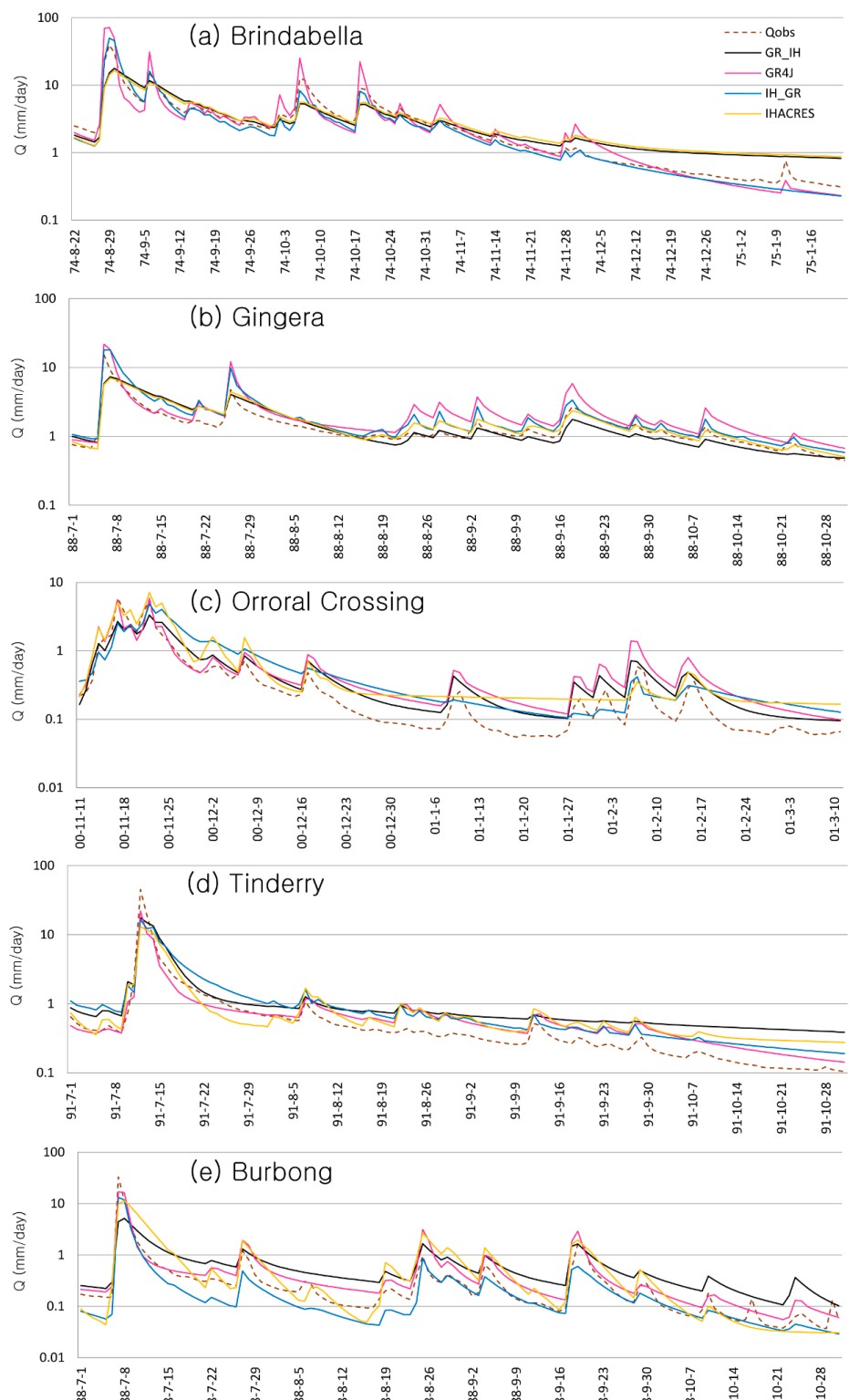

**Figure 4.** Comparison of the hydrographs of the four rainfall–runoff models for the maximum events with respect to the NSElog objective function: (**a**) validation of the 1970s with the calibration of the 2000s for the Brindabella (wet) catchment, (**b**) validation of the 1980s with the calibration of the 1970s for the Gingera (wet) catchment, (**c**) validation of the 2000s with the calibration of the 1980s for the Orroral Crossing (intermediate) catchment, (**d**) validation of the 1990s with the calibration of the 1970s for the Tinderry (dry) catchment, and (**e**) validation of the 1980s with the calibration of the 1990s for the Burbong (dry) catchment.



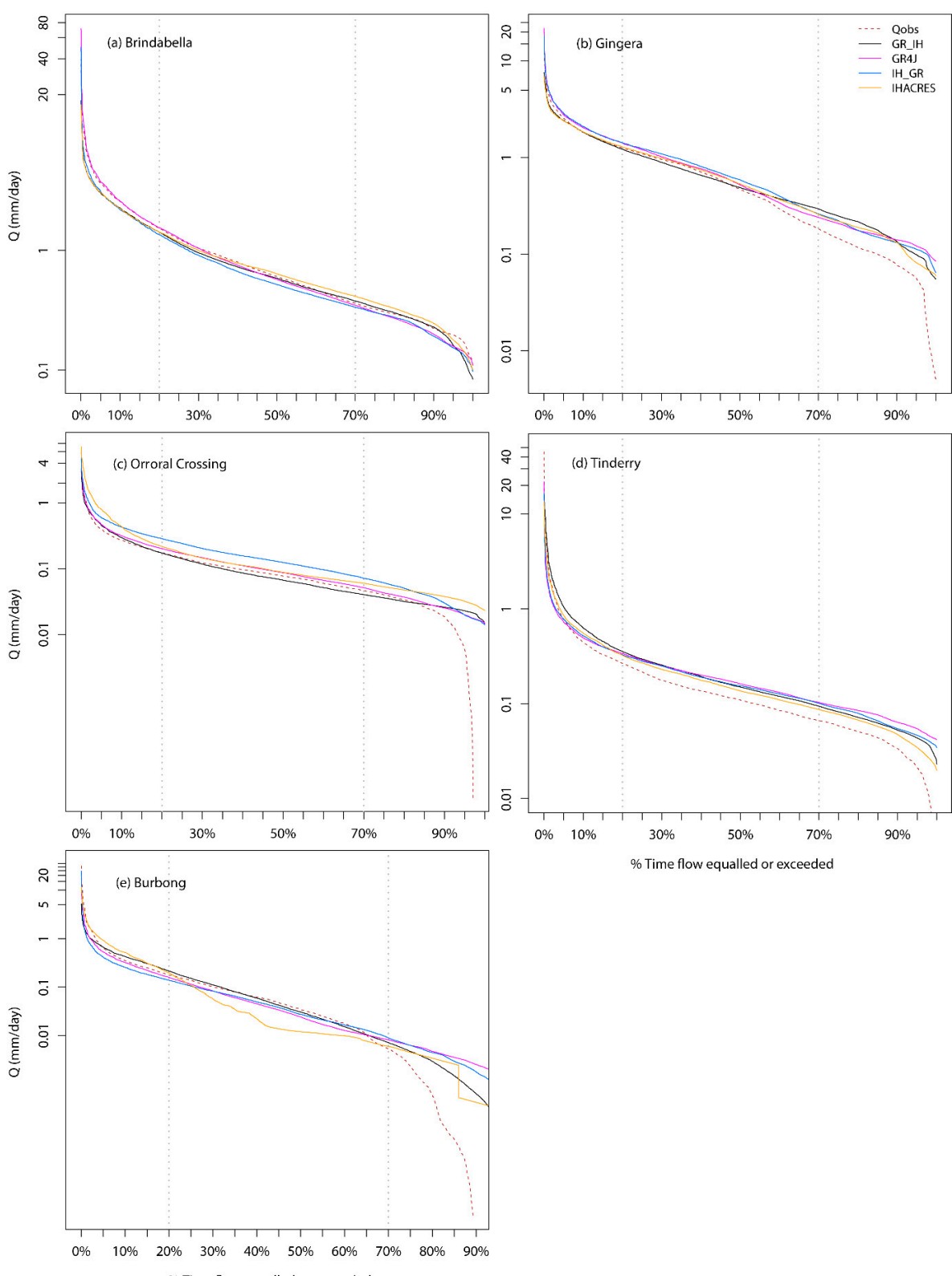

**Figure 5.** Comparison of the flow duration curves of the four rainfall–runoff models with respect to the NSElog objective function: (**a**) validation of the 1970s with the calibration of the 2000s for the Brindabella (wet) catchment, (**b**) validation of the 1980s with the calibration of the 1970s for the Gingera (wet) catchment, (**c**) validation of the 2000s with the calibration of the 1980s for the Orroral Crossing (intermediate) catchment, (**d**) validation of the 1990s with the calibration of the 1970s for the Tinderry (dry) catchment, and (**e**) validation of the 1980s with the calibration of the 1990s for the Burbong (dry) catchment.

The IH_GR model had similar or lower NSElog values than the GR4J model, whereas its NSElog values were similar or higher than those of the IHACRES model (Figure 3). This means that the production component of the IHACRES model does not improve the performance of the GR4J model, whereas the routing component of the GR4J model improves the performance of the IHACRES model with respect to the low-flow simulation. The hydrograph of the IH_GR model in Figure 4a confirms the better simulation result compared to the IHACRES model. This may be due to use of the power function for routing in the GR4J model, which generally simulates a recession flow better than the exponential function used in the IHACRES model.

4.2.2. Intermediate Catchment

The NSElog values of the 2000s of the GR_IH model for the Orroral Crossing catchment were higher than those of the GR4J model, regardless of the calibration and validation purposes (Figure 3). The routing component of the IHACRES model therefore improves the GR4J model for this period (Figure 4c). Figure 5c also complements the result by showing the better simulations of the GR_IH model for the recession and low flow (15–30%). This was not an expected result because the routing component of the IHACRES model uses an exponential function, which generally produces less mild recession flow than the power function that is used in the GR4J model. In addition, the GR_IH model had much higher NSElog values than the IHACRES model (Figure 3); therefore, the production component of the GR4J model improves the IHACRES model in terms of the recession- and low-flow simulation.

The IH_GR model had lower NSElog values than the GR4J model (Figure 3); therefore, the production component of the IHACRES model does not improve the performance of the GR4J model. The flow duration curve of the IH_GR model is higher than that of the GR4J model over nearly all periods, which implies that the production component of the IHACRES model produces significantly larger effective rainfall and therefore causes the poor performance. In addition, the IH_GR model had NSElog values as low as that of the IHACRES model; therefore, the routing component of the GR4J model does not help to improve the performance of the IHACRES model.

4.2.3. Dry Catchments

The NSElog values of the GR_IH model for the dry Tinderry catchment were higher than those of the GR4J model for the calibration and validation periods with the calibration of the 2000s (Figure 3). Therefore, the routing component of the IHACRES model improves the GR4J model in terms of the recession- and low-flow simulation of this period, although the performance of the two models is similar for the other calibration periods. This implies that the power function in the routing component of the GR4J model is not always better than the exponential function in the routing component of the IHACRES model for the recession- and low-flow simulations for the dry catchment. In contrast, the IH_GR model had lower NSElog values than the GR4J and IHACRES models; therefore, the production component of the IHACRES model and routing component of the GR4J model do not improve the performances of the GR4J and IHACRES models, respectively. Figure 4d complements these results because it shows the relatively poor recession hydrograph of the IH_GR model. Figure 5d also shows the relatively poor recession- and low-flow simulation by the IH_GR model.

The GR_IH and IH_GR models of the Burbong catchment had higher NSElog values than the IHACRES model, and the GR_IH model had higher NSElog values than the IH_GR model. Figure 5e shows that the recession- and low-flow simulation of the GR_IH model is closer to the observed flow than that of the IH_GR model. This means that the production component of the GR4J model is more important for the low-flow simulation of the very dry Burbong catchment than the routing component of the GR4J model for improving the rainfall–runoff model. The recession hydrographs in Figure 4e show a steeper recession for the IHACRES model, whereas the GR_IH and IH_GR models have a mild recession, which

is similar to the shape of the observed recession hydrograph. The production and routing components of the GR4J model are therefore important for the low-flow simulation of the very dry catchment with respect to the model improvement.

For the results based on the 2000s calibration for the Burbong catchment, the NSElog values of all rainfall–runoff models were not good for the calibration period and the NSElog values of the validation periods were poor. This means that all models were unsuitable to simulate the 2000s period, which includes the millennium drought [16], and the calibrated parameter values could not properly simulate the remaining validation periods. Therefore, it is necessary to improve the rainfall–runoff model to simulate the millennium drought of a very dry catchment such as the Burbong catchment, but it was beyond the scope of this study and will be covered in future study.

*4.3. Sensitivity Analysis of Parameters of Rainfall–Runoff Models*

Figure 6 shows the sensitivity of the parameters of the four rainfall–runoff models to the NSE* target function, which focuses on the parameter sensitivity to high flow. The numbers in the figure represent the numbers of the parameters in Table 2. For some parameters, however, we used the name of the parameter instead of the number for clarity. The horizontal and vertical axes represent the maximum and minimum TSI values, respectively, among the four TSIs of the four decadal periods. The red line is the one-to-one line of the TSI value. If the TSI value of a parameter is near or on the one-to-one line then it means that the parameter has the same sensitivity over all data periods used because maximum and minimum TSI values are equal.

Parameter no. 4 ($x4$) for the unit hydrograph of the GR4J model became more sensitive when it was coupled with the production component of the IHACRES model (i.e., the IH_GR model). For example, parameter no. 4 of the IH_GR model for the Gingera catchment (Figure 6d) had a higher maximum TSI value than that of the GR4J model (Figure 6a). This means that parameter no. 1 ($f$) of the IHACRES model makes parameter no. 4 more sensitive. However, the sensitivity of parameters no. 2 ($x2$) and no. 3 ($x3$) of the IH_GR model (Figure 6d) was reduced compared to the GR4J model (Figure 6a), which could be due to compensation of the sensitivity between parameters no. 2, no. 3, and no. 4.

Parameter no. 2 ($\tau_s$) of the GR_IH model (Figure 6c) was insensitive when the target function applied is the NSE* for high-flow sensitivity. This represents proper functioning because this parameter is related to the low-flow simulation. Parameters no. 3 ($\tau_q$) and no. 4 ($v_s$) of the GR_IH model (Figure 6c) were closer to the one-to-one line than those of the IHACRES model (Figure 6b). This means that parameter no. 1 ($x1$) of the GR4J model reduces the gap between the maximum and minimum parameter sensitivity during the four periods for the respective parameters no. 3 ($\tau_q$) and no. 4 ($v_s$) of the IHACRES model. In other words, parameter no. 1 ($x1$) has the problem of reducing the difference of sensitivities according to the different characteristics of data periods such as wet and dry periods for parameters no. 3 ($\tau_q$) and no. 4 ($v_s$).

Figure 7 shows the sensitivity of the parameters of the four rainfall–runoff models to the NSElog* target function, which focuses on the parameter sensitivity to the low flow. Parameter no. 4 ($x4$) for the time base of the unit hydrograph of the IH_GR model was insensitive to the NSElog* target function (Figure 7e). The insensitivity of this parameter to the NSElog* target function represents proper functioning because the time base is more sensitive for capturing high flow [5]. Parameters no. 2 ($x2$) and no. 3 ($x3$) of the IH_GR model had a reduced sensitivity to the NSElog* target function (Figure 7e) compared to those of the GR4J model (Figure 7a) because of the influence of parameter no. 1 ($f$) of the IHACRES model. However, these IH_GR model parameters were still sensitive because they had sensitivities higher than a sensitivity threshold of 0.2 in terms of the maximum TSI.

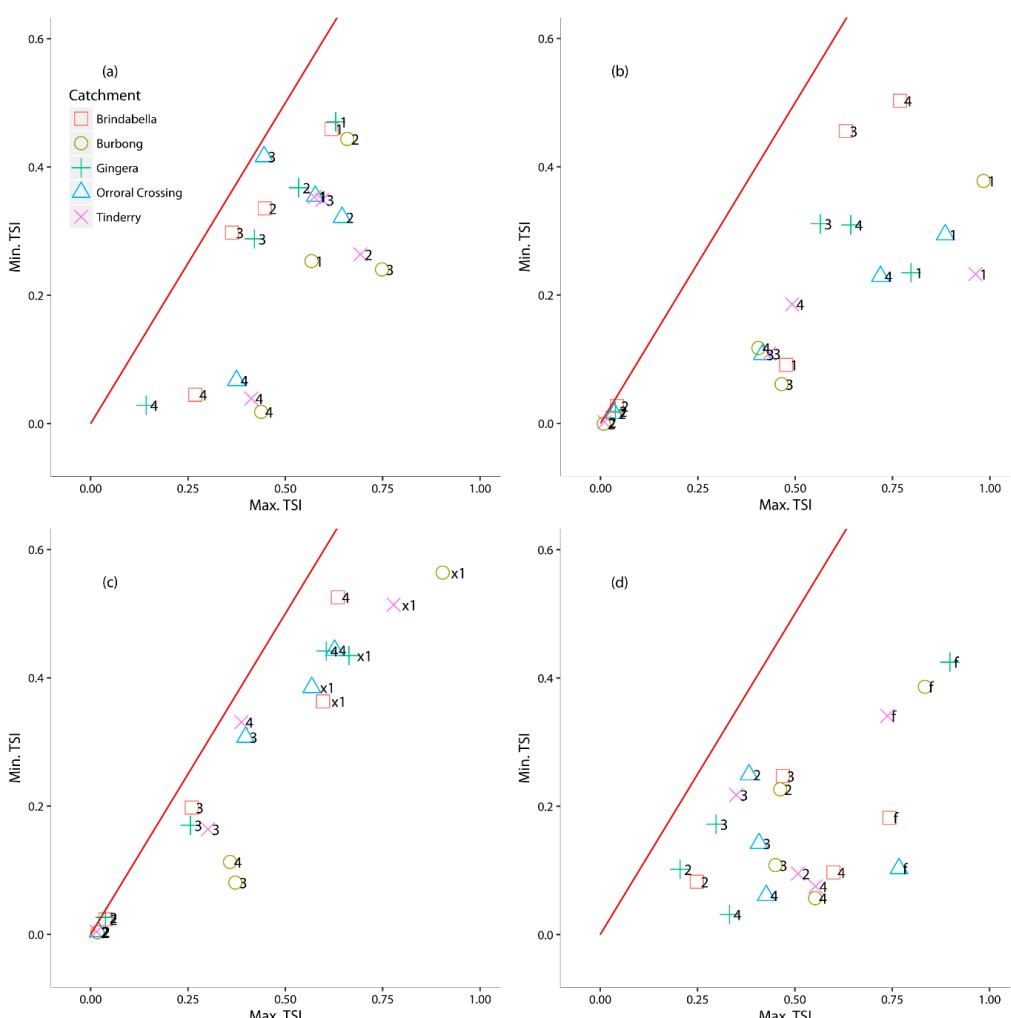

**Figure 6.** Sensitivity of the parameters of the four rainfall–runoff models for the five catchments to the NSE* target function:
(**a**) result of the GR4J model, (**b**) IHACRES model, (**c**) GR_IH model (*x*1 is the parameter of the GR4J model), and (**d**) IH_GR
model (*f* is the parameter of the IHACRES model). The one-to-one line is shown in red.

The maximum TSI value of parameter no. 2 ($\tau_s$) of the GR_IH model for the NSElog*
target function (Figure 7d) is overall higher than that of the IHACRES model (Figure 6b,c).
This indicates that parameter no. 1 (*x*1) of the GR4J model makes parameter no. 2 ($\tau_s$) of
the IHACRES model, related to the low-flow simulation, more suitable for the low-flow
simulation. However, parameter no. 1 (*x*1) of the GR_IH model makes parameter no. 3 ($\tau_q$)
more sensitive to the NSElog* target function (Figure 7d) compared to the IHACRES model
(Figure 7c), even though parameter no. 3 ($\tau_q$) is for the high-flow simulation. This means
that the combination of parameter no. 1 (*x*1) of the GR4J model and parameter no. 3 ($\tau_q$)
of the IHACRES model is not suitable in terms of the model structure. This unsuitability
of the GR_IH model structure could have caused the poor performance of the low-flow
simulation for the wet catchments, as discussed in Section 4.2.

The components that improve the model performance are summarized in Table 3.
For example, if the aim is to improve high-flow simulation for the Brindabella catchment,
a helpful production component is IH (see the fourth column in Table 3), which is the
production store of the IHACRES model. This result was derived through the comparison of
the original GR4J model and the IH_GR model, as described in Section 4.1.1. The rest of the
results in the fourth and fifth columns are also the results obtained in Sections 4.1 and 4.2
through this kind of comparison. Lastly, the results in the sixth column show the best
models when all the analysis results in Section 4 are comprehensively considered.

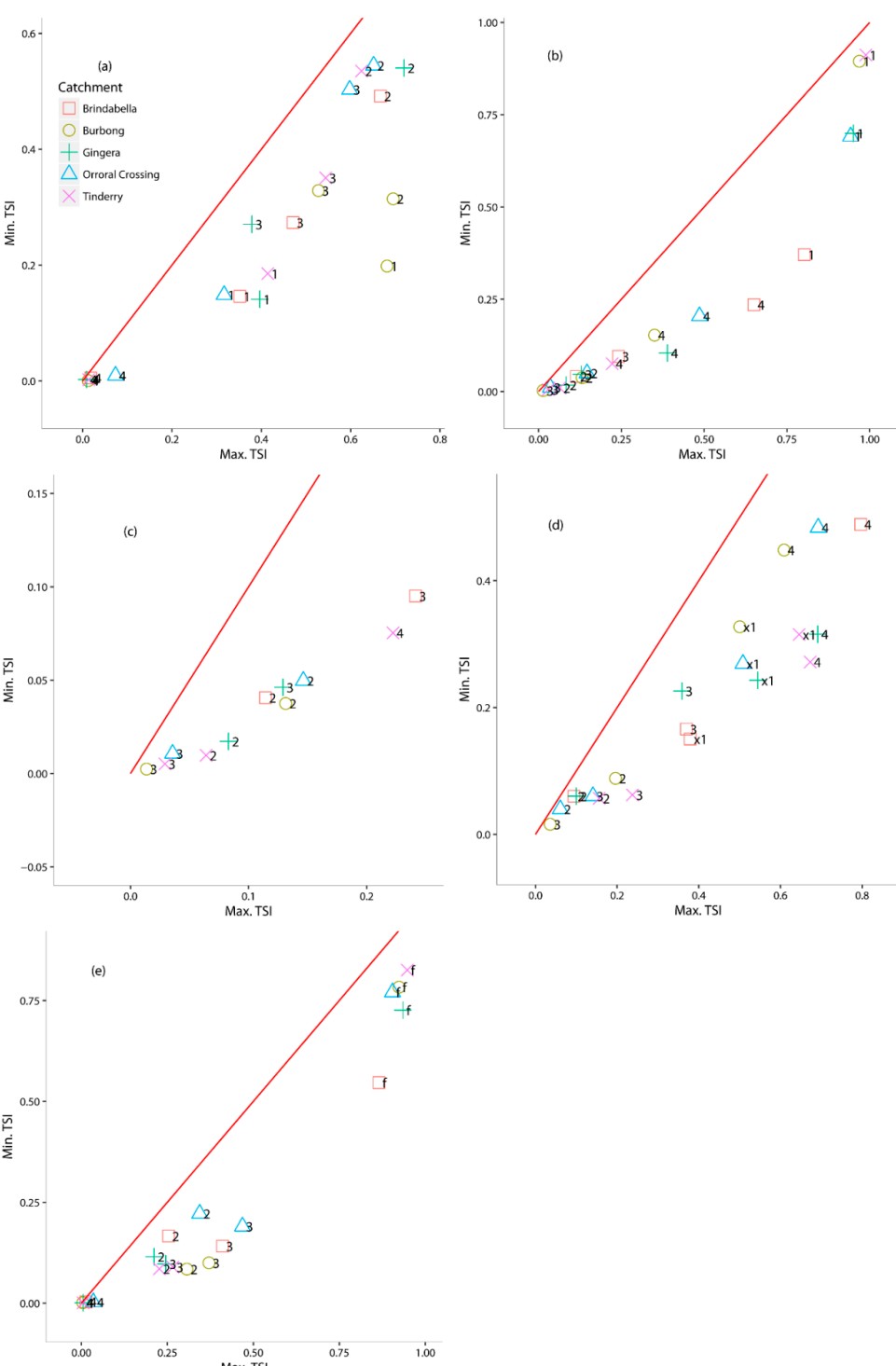

**Figure 7.** Sensitivity of the parameters of the four rainfall–runoff models for the five catchments to the NSElog* target function: (**a**) result of the GR4J model, (**b**) IHACRES model, (**c**) zoom of (**b**), (**d**) GR_IH model (*x*1 is the parameter of the GR4J model), and (**e**) IH_GR model (*f* is the parameter of the IHACRES model). The one-to-one line is shown in red.

**Table 3.** The components of the GR4J and IHACRES models that improve the original IHACRES or GR4J model performances with respect to the aim of the simulation and the characteristics of the catchments.

| Aim | Catchment Characteristics | Catchment | Production Component | Routing Component | Best Model [a] |
|---|---|---|---|---|---|
| Improvement of high-flow simulation | Wet | Brindabella | IH | IH | IHACRES |
| | | Gingera | IH | IH | IHACRES |
| | Intermediate | Orroral Crossing | GR | - | GR_IH |
| | Dry | Tinderry | - | GR | GR4J |
| | | Burbong | - | GR | IH_GR |
| Improvement of low-flow simulation | Wet | Brindabella | - | GR | GR4J |
| | | Gingera | - | GR | GR4J |
| | Intermediate | Orroral Crossing | GR | IH | GR_IH |
| | Dry | Tinderry | - | IH | GR_IH |
| | | Burbong | GR | GR | GR4J |

[a] Among the best models, the GR_IH model is the combination of the production store of the GR4J model and the routing store of the IHACRES model (vice versa in the IH_GR model).

## 5. Conclusions

In this study, the improvement of performances for the GR4J and IHACRES models was investigated by component combination and sensitivity analysis. We investigated the pros and cons of the components of the rainfall–runoff models using the five catchments and four 10-year periods of data that have different climatic characteristics. For the high-flow simulation, the components of the IHACRES model improve the model performance for wet catchments, whereas the components of the GR4J model improve the model performance for intermediate and dry catchments. For the low-flow simulation, the components of the GR4J model improve the model performance for most catchments, whereas the components of the IHACRES model improve the model performance for two catchments, Orroral Crossing and Tinderry. The components of the IHACRES model are therefore appropriate for the high-flow simulation of wet catchments, whereas the components of the GR4J model are suitable for the low-flow simulation. However, the routing component of the IHACRES model is sometimes useful for the improvement of the low-flow simulation for intermediate and dry catchments. In addition, considering these results comprehensively, the IHACRES model is suitable for high-flow simulation of wet catchments and the GR4J model is suitable for simulating the runoff of intermediate and dry catchments.

The suitability of the parameters of the two new models generated by the component combination method was investigated by sensitivity analysis. In addition, the interaction between parameters and the reasons for poor model performance were examined by sensitivity analysis. The results of this study can be useful for future improvement of the GR4J and IHACRES models. In addition, the methodology used in this study can be applied to the improvement of other rainfall–runoff models of interest.

As a limitation of this study, all rainfall–runoff models could not adequately simulate the millennium drought in the Burbong catchment, which is a dry catchment. The improvement of the rainfall–runoff model to adequately simulate the millennium drought was outside the scope of this study and will be carried out in future studies.

**Author Contributions:** Conceptualization, M.-J.S. and C.-S.K.; Methodology, M.-J.S.; Software, M.-J.S.; Validation, M.-J.S. and C.-S.K.; Formal Analysis, M.-J.S. and C.-S.K.; Investigation, M.-J.S. and C.-S.K.; Resources, M.-J.S.; Data Curation, M.-J.S.; Writing-Original Draft Preparation, M.-J.S.; Writing-Review and Editing, C.-S.K.; Visualization, M.-J.S.; Supervision, C.-S.K. All authors have read and agreed to the published version of the manuscript.

**Funding:** This research received no external funding.

**Institutional Review Board Statement:** Not applicable.

**Informed Consent Statement:** Not applicable.

**Data Availability Statement:** The data are available on the request of the corresponding author.

**Acknowledgments:** We gratefully acknowledge the ACTEW Corporation for supplying the rainfall data for the five study catchments and streamflow data for the Gingera and Tinderry catchments. We also gratefully acknowledge the ACT and NSW governments for providing the streamflow data for the Burbong and Orroral Crossing catchments and streamflow data for the Brindabella catchment, respectively.

**Conflicts of Interest:** The authors declare no conflict of interest.

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
