# Peer review of "Component Combination Test to Investigate Improvement of the IHACRES and GR4J Rainfall–Runoff Models"

_water, doi:10.3390/w13152126_

Round 1

Reviewer 1 Report

Authors need to revise Abstract to give clearer findings obtained through this study.  The terms ‘production’ and ‘both components of the GR4J’ is not clear for me or potential readers of this manuscript, if accepted. 

In introduction section, authors did not provide necessity of this study clearly.  Authors need to revise or add more details to provide why authors started this study (Lines 53-54). Two models, GR4J and IHACRES, were introduced here without any explanation of model structure/components, thus very hard to understand novelty of this study.  Please reorganize the Introduction section to provide why this study is important.  

In Line 94, authors described daily rainfall and PET are input for the models.  Are these only input to GR4J and IHACRES models? Authors need to add additional paragraphs to describe model structure and input parameters of these two models. 

Did authors calibrate tow rainfall-runoff models using flow data for flow regime?  Or did authors calibrate model for high flow and low flow ranges, respectively. ( I am asking this question because high and low flow prediction using two models, described in Introduction and methodology section).

In Lines 166 and 182, the term ‘production’ is used.  This is not common term in hydrology and rainfall-runoff simulation models.  Please describe this term in details. 

Did author calibrate model for quick and slow flows separately?  If SCE global optimization algorithm using all flow regime data together was used, these quick and slow flows are not considered. 

As I read through the manuscript, I was able to find how two model components (production store from GR4J and routing store of IHACRES model – this is GR_IH and vice versa in IH_GR model) were incorporated together.  These 2-3 lines of explanation would be helpful in understanding the Abstract. 

As I read sections 4.1.1, I have one question.  What will happen if authors optimize model parameter for high flow observation data using IHACRES or GR_IH model or IH_GR model ( this is for wet catchment)?  And also for intermediate catchment using other appropriate model separately. 

Also if authors use the ‘high flow’ and ‘low flow’ term, it would be very difficult for potential users of this study when they apply authors’ results in their own watershed.  Can authors provide any clearer standard for high flow (or wet catchment) and intermediate/dry catchment also ?

In Figure 5, model predictions (4 model) did not match for low flow regime in general.  Do author provide any reasons for these mismatches?

In Conclusion section, authors provide the results obtained in this study,  I am wondering whether authors can provide any single rainfall-runoff model, which is suitable for prediction of flow for flow regime (wet, intermediate, and dry catchment) based on the results obtained from this study.  Most model users want to have one single model, instead of 4 (or 2), to predict flow.  I think if authors provide these in the Conclusion section, it will be very helpful research article, if accepted after review processes.

Reviewer 2 Report

The study is poorly justified and it is clear from the conclusions that its adds little to science and to the practical applications of rainfall-runoff models (l.509-510): "The results of this study might be useful for future improvement of the GR4J and IHACRES models." 

(1) The abstract is unclear. Please Introduce "the components" and the number of wet, intermediate and dry catchments tested. Improved (l.20, 21) relative to what?

(2) The justification for the study presented in the first paragraph is based on just five articles, two of which are written by you (first author). No short summary is given of the content of these paper with  a clear explanation how the current paper is different. All kind of sensitivity analyses were already done by you in ref (2), so why does it need to be done again in this paper? 

(3) The IHACRES and GR4J models have been applied in many studies, but the following statement only refers to your (first author) own papers: "the IHACRES model is strong for high-flow simulation, whereas the GR4J model is strong for low-flow simulation." Please explain these findings and add a concise a summary of the global scientific literature. 

(4) What is the point of having a model for high-flow simulation and a model for low-flow simulation?! A model should be able to cover both flow conditions. Please present an multi-objective function optimization or use the average of the NSE and the NSE-log.  

(5)  Please present a clear summary table with the results of the NSE optimizations, add the observed rainfall and runoff of the watersheds for the modelled decades in the table and reduce the text, which is tedious to read.

(6) What is the added value of the sensitivity analysis?! The results of the NSE optimizations already show which are the best models for the specific cases and all parameters have been optimized. The whole sensitivity analysis can be deleted. 

(7) The language of the article and Table 2 are confusing. The study is an optimization of four rainfall-runoff models, so "improve" relative to what? 

(8) It is unclear why you select to optimize the production store of the GR4J model, while you keep the production store for the IHACRES model constant. Your justification is a reference to the article of Croke and Jakeman, where we find the following: "the optimum value of d is 200 mm or greater."  And in Croke and Jakeman (J. Hydrology, 2006), we find values of d between 220 and 590 mm. Please redo the simulations with the production store as an optimizable parameter. 

(9) Please explain the changes in land and water use in the five catchments over the 40 year period and how this is considered in the modeling optimization. 

Round 2

Reviewer 1 Report

Authors revised the manuscript based on 1st review comments.

I think authors may need to add limitations of this study in the Conclusion section.

Reviewer 2 Report

See previous comments and responses.

(2), (3) Please provide concise summaries of the literature.

Ad (3) Identify studies that report that GR4J is not good for high flow and IHACRES is not good for low flow.

(4) These models have been around for long time. Not sufficient to write a paper just to analyze model components for improvement.

(5) Please provide a summary table with rainfall and runoff data for the 5 catchments and 4 decades, so the reader can understand the performance of the models!

(7) Table 2 remains unclear. Please tabulate which model performs best for each catchment and case (prod-routing) : GR4J-GR4J  GR4J-IH  IH-GR4J IH-IH.

(8) Higher values were found in Croke and Jakeman (J. Hydrology, 2006)! So please present some simulation with a variable production store to prove that there is no improvement 

(9) Please reflect on the changes in land use on the runoff and the model parameters.

Round 3

Reviewer 2 Report

Dear Authors,

Thanks for the corrections and additions, it is much better now. I have one more small comment. 

The addition of the best-model column in Table 3 (previously Table 2) is very helpful but as a whole the Table remains impossible to understand.

Improvement of high flow - original model is IHACRES? 

But then how can we have:  IH IH IH-IH ?  This means the original model was GR-GR ?

And how can we have:  -  GR  GR-GR ? This means here the original model was GR-IH?!

Maybe just give two data columns (1) original model (2) final best model ?

Explain the choice of selection of the original model in the text and explain in the caption or table headers or foot note that X-X means Production-Routing.
